# YKL40/Integrin β4 Axis Induced by the Interaction between Cancer Cells and Tumor-Associated Macrophages Is Involved in the Progression of High-Grade Serous Ovarian Carcinoma

**DOI:** 10.3390/ijms251910598

**Published:** 2024-10-01

**Authors:** Keitaro Yamanaka, Yu-ichiro Koma, Satoshi Urakami, Ryosuke Takahashi, Satoshi Nagamata, Masaki Omori, Rikuya Torigoe, Hiroki Yokoo, Takashi Nakanishi, Nobuaki Ishihara, Shuichi Tsukamoto, Takayuki Kodama, Mari Nishio, Manabu Shigeoka, Hiroshi Yokozaki, Yoshito Terai

**Affiliations:** 1Division of Pathology, Department of Pathology, Kobe University Graduate School of Medicine, Kobe 650-0017, Japan; 213m889m@gsuite.kobe-u.ac.jp (K.Y.); urasato@med.kobe-u.ac.jp (S.U.); 239m863m@gsuite.kobe-u.ac.jp (M.O.); rikuarw@med.kobe-u.ac.jp (R.T.); h450@med.kobe-u.ac.jp (H.Y.); 215m857m@stu.kobe-u.ac.jp (T.N.); n.ishihara@people.kobe-u.ac.jp (N.I.); stsuka@med.kobe-u.ac.jp (S.T.); takodama@med.kobe-u.ac.jp (T.K.); marin@med.kobe-u.ac.jp (M.N.); mshige@med.kobe-u.ac.jp (M.S.); hyoko@med.kobe-u.ac.jp (H.Y.); 2Division of Obstetrics and Gynecology, Department of Surgery Related, Kobe University Graduate School of Medicine, Kobe 650-0017, Japan; takaryo@med.kobe-u.ac.jp (R.T.); nagamata@med.kobe-u.ac.jp (S.N.); yterai@med.kobe-u.ac.jp (Y.T.); 3Division of Gastroenterology, Department of Internal Medicine, Kobe University Graduate School of Medicine, Kobe 650-0017, Japan; 4Division of Hepato-Biliary-Pancreatic Surgery, Department of Surgery, Kobe University Graduate School of Medicine, Kobe, 650-0017, Japan; 5Division of Gastrointestinal Surgery, Department of Surgery, Kobe University Graduate School of Medicine, Kobe, 650-0017, Japan

**Keywords:** epithelial ovarian cancer, high-grade serous ovarian carcinoma, tumor-associated macrophages, tumor microenvironment, YKL40

## Abstract

Macrophages in the tumor microenvironment, termed tumor-associated macrophages (TAMs), promote the progression of various cancer types. However, many mechanisms related to tumor–stromal interactions in epithelial ovarian cancer (EOC) progression remain unclear. High-grade serous ovarian carcinoma (HGSOC) is the most malignant EOC subtype. Herein, immunohistochemistry was performed on 65 HGSOC tissue samples, revealing that patients with a higher infiltration of CD68^+^, CD163^+^, and CD204^+^ macrophages had a poorer prognosis. We subsequently established an indirect co-culture system between macrophages and EOC cells, including HGSOC cells. The co-cultured macrophages showed increased expression of the TAM markers CD163 and CD204, and the co-cultured EOC cells exhibited enhanced proliferation, migration, and invasion. Cytokine array analysis revealed higher YKL40 secretion in the indirect co-culture system. The addition of YKL40 increased proliferation, migration, and invasion via extracellular signal-regulated kinase (Erk) signaling in EOC cells. The knockdown of integrin β4, one of the YKL40 receptors, suppressed YKL40-induced proliferation, migration, and invasion, as well as Erk phosphorylation in some EOC cells. Database analysis showed that high-level expression of YKL40 and integrin β4 correlated with a poor prognosis in patients with serous ovarian carcinoma. Therefore, the YKL40/integrin β4 axis may play a role in ovarian cancer progression.

## 1. Introduction

Ovarian cancer is the most lethal gynecologic malignancy, with a persistently high mortality rate [1]. Epithelial ovarian cancer (EOC) constitutes 95% of ovarian cancers, with high-grade serous ovarian carcinoma (HGSOC) accounting for 70% of EOC cases [2]. The initial treatment for EOC, including HGSOC, involves cytoreductive surgery and platinum-based chemotherapy. In recent years, maintenance therapy has significantly improved with the use of anti-angiogenic agents such as bevacizumab and poly (adenosine diphosphate–ribose) polymerase (PARP) inhibitors, which are involved in DNA damage repair processes [3]. However, over 80% of cases are diagnosed at an advanced stage—when the tumor has spread to lymph nodes and the upper abdominal organs—resulting in a 5-year survival rate of approximately 30% [4]. The high mortality rate is significantly influenced by the chemoresistance of ovarian cancer [5]. Therefore, elucidating the mechanism of HGSOC progression is important for the control of ovarian cancer.

The tumor microenvironment, comprising cancer cells and stroma, contributes to cancer progression through mutual interactions [6]. HGSOC is characterized by a unique pattern of dissemination in the peritoneal space, where interactions between cancer and stromal cells contribute to disease progression. The stromal cells in the tumor microenvironment include macrophages, fibroblasts, mesothelial cells that constitute the peritoneum, adipocytes abundant in the omentum, cytotoxic T cells, and myeloid-derived suppressor cells [7]. For instance, fibroblasts, referred to as cancer-associated fibroblasts, secrete transforming growth factor (TGF)-β, a well-known humoral factor for EOC progression. The combination therapy of TGF-β inhibitors and cisplatin significantly inhibits the growth of cisplatin-resistant ovarian xenografts [8]. CD93, also known as complement protein 1 q subcomponent receptor, which is expressed on monocytes, neutrophils, B cells, and NK cells, has been reported to be involved in the regulation of angiogenesis for tumor growth [9]. Additionally, EOC cells undergo epithelial-mesenchymal transition, upregulating integrin α5 to adhere to mesothelial cells and facilitate dissemination from the primary tumor [10]. The presence of CD8^+^ tumor-infiltrating lymphocytes is strongly associated with a favorable prognosis in patients with ovarian cancer [11]. Increased programmed death (PD)-ligand 1 expression in ovarian tumors correlates with a reduction in intra-tumoral CD8^+^ lymphocytes and poorer patient survival [12]. In a single-arm Phase 1 and 2 trial of niraparib (a PARP inhibitor) in combination with pembrolizumab (an anti-PD-1 antibody) for patients with recurrent platinum-resistant ovarian cancer, the disease control rate was 65% [13]. Considering the extensive research, the analysis of tumor–stromal interactions in EOC progression is important, although many of the related mechanisms remain unclear.

Macrophages, known as tumor-associated macrophages (TAMs) in the tumor microenvironment, are one of the most abundant immune cell types within the tumor [14]. A recent study utilizing single-cell omics has highlighted the diversity of macrophages [15]. They are broadly categorized into tumor-suppressive (M1) and tumor-promoting (M2) types, and TAMs have an M2-like phenotype [16]. The scavenger receptors CD163 and CD204 are commonly used as TAM markers [17]. In ovarian cancer, data from the Cancer Genome Atlas Program indicate that macrophages are the most prevalent immune cell type, with 51% being M2 macrophages [18]. This underscores the critical role of TAMs in ovarian cancer. Therefore, this study focused on the interaction between cancer cells and macrophages in HGSOC progression.

Herein, the authors established an in vitro co-culture system using immortalized EOC cell lines and peripheral blood-derived macrophages. The major signaling pathways involving cytokines and their receptors that induce the proliferation, migration, and invasion of EOC cell lines were investigated. Furthermore, the clinical significance of these cytokines and their receptors was examined.

## 2. Results

### 2.1. Number of Infiltrating Macrophages Correlates with Poor Prognosis in Ovarian Cancer

The authors conducted an immunohistochemical examination to investigate the relationship between macrophages and clinical data in surgical specimens of HGSOC (Figure 1A). In the Kaplan–Meier analysis, the group with a higher infiltration of pan-macrophage marker CD68^+^ and TAM marker CD163^+^ exhibited poorer overall survival (OS) and progression-free survival (PFS). Additionally, higher infiltration levels of TAM marker CD204^+^ were associated with poorer OS (Figure 1B,C). Regarding clinicopathological indicators, higher infiltration levels of any of the markers were significantly associated with chemotherapy resistance. Furthermore, the infiltration of CD68^+^ macrophages was significantly correlated with lymph node metastasis (Table 1). These results suggest that macrophages may be involved in the progression of ovarian cancer.

### 2.2. Co-Culture of Macrophages and EOC Cells Promotes Polarization of Macrophages into TAMs and Enhances Malignant Phenotypes of EOC Cells

Considering the correlation between macrophages and HGSOC prognosis, the authors focused on the interaction between macrophages and EOC cells. An indirect co-culture system using macrophages derived from peripheral blood and EOC cell lines was established; these cells were also monocultured and used as a control (Figure 2A). To determine whether macrophages polarize into TAMs through indirect co-culture, real-time quantitative polymerase chain reaction (qPCR) for TAM markers was performed. The expression level of TAM markers CD163 and CD204 in macrophages was elevated in co-cultures with all EOC cells (Figure 2B). Next, the authors investigated which signal pathways are activated in EOC cells through co-culture and how these signals affect malignant phenotypes. Western blotting revealed Erk activation in all co-cultured EOC cells compared with monocultured EOC cells (Figure 2C and Appendix A). Furthermore, the MTS assay revealed a significant increase in the proliferation of co-cultured EOC cells, although the effect was modest (Figure 2D). Transwell migration and invasion assays demonstrated a significant enhancement of migration and invasion abilities in the co-cultured EOC cells, respectively (Figure 2E,F and Appendix A).

### 2.3. Increased YKL40 Expression Resulting from Indirect Co-Culture of Macrophages and EOC Cells Promotes Malignant Phenotypes of EOC Cells

To investigate which humoral factors affect the proliferation, migration, and invasion of EOC cells in the indirect co-culture system, the authors collected culture supernatants from EOC cell monocultures, macrophage monocultures, and EOC cell/macrophage co-cultures (Figure 3A). The culture supernatants from experiments using KURAMOCHI cells were applied to cytokine arrays, and the results revealed an increased expression of YKL40, osteopontin (OPN), interleukin (IL)-8, CCL2, IL-17A, and matrix metalloproteinase (MMP) 9 in the supernatants of the KURAMOCHI/macrophage co-culture compared with the KURAMOCHI and macrophage monocultures (Figure 3B and Appendix A). Additionally, OPN, IL-8, CCL2, and MMP9 were the most abundantly secreted in the supernatants of the KURAMOCHI/macrophage co-culture, confirmed using enzyme-linked immunosorbent assay (ELISA) (Appendix A). Although many studies have reported these factors in EOC, this study focused on YKL40—which has been relatively less reported—to determine a novel mechanism in EOC progression. Using ELISA, YKL40 secretion was confirmed in three EOC cell lines and its levels were elevated in the supernatants of the EOC cell/macrophage co-culture in all cell lines (Figure 3C). Notably, in EOC cell lines other than KURAMOCHI, secretion from the EOC cells was minimal, and secretion from macrophages was higher than that from the EOC cells (Figure 3C). To determine whether YKL40 contributes to the malignant phenotypes of EOC cells, recombinant human YKL40 (rhYKL40) was added to the EOC cells. When rhYKL40 was added at 250 ng/mL, there was a slight but significant increase in proliferation compared with that in EOC cells without rhYKL40. However, increasing the concentration to 500 ng/mL did not result in a consistent dose-dependent increase in proliferation (Figure 3D). Furthermore, the addition of rhYKL40 enhanced the migration and invasion of EOC cells, where a concentration of 250 ng/mL was sufficient (Figure 3E,F and Appendix A).

### 2.4. YKL40 Promotes Malignant Phenotypes of EOC Cells via Erk Pathway

The authors investigated the signaling pathways through which YKL40 affects the malignant phenotypes of EOC cells. EOC cells were treated with rhYKL40, and the Erk pathway was evaluated using Western blotting at 0, 10, 30, and 60 min. In all three EOC cell lines, phosphorylation of Erk was consistently observed at 10 min (Figure 4A and Appendix A). Additionally, treatment with MEK1/2 inhibitor (PD98059) significantly suppressed the enhancements in proliferation, migration and invasion induced by rhYKL40 (Figure 4B–D and Appendix A).

### 2.5. Enhanced Effects of Malignant Phenotypes of EOC Cells by YKL40 Are Mediated through Integrin β4

Our previous research identified integrin β4 (ITGB4) as a novel receptor for YKL40 [19]. Therefore, we focused on ITGB4 and investigated its impact on EOC cells. Specifically, the authors evaluated how YKL40 affects malignant phenotypes and the Erk pathway via ITGB4. Using RNA interference, ITGB4 expression in EOC cells was suppressed, and the successful silencing of mRNA and protein levels was confirmed (Figure 5A,B and Appendix A). In all three EOC cell lines, rhYKL40-induced proliferation was inhibited by ITGB4 knockdown (Figure 5C). Regarding migration and invasion, the enhancing effects of rhYKL40 were suppressed by ITGB4 knockdown in all three EOC cell lines, whereas in the SKOV3 cell line, migration and invasion were enhanced in the absence of rhYKL40 (Figure 5D,E and Appendix A). In the Erk pathway, p-Erk levels were significantly reduced via ITGB4 knockdown at 10 min in the KURAMOCHI and OVCAR3 cell lines but not in the SKOV3 cell line (Figure 5F and Appendix A).

### 2.6. YKL40/ITGB4 Axis May Act as Indicator of Poor Prognosis in EOC

To validate the results of the in vitro experiments, the authors performed immunohistochemical analysis of YKL40 and ITGB4 using 65 human HGSOC tissues (Figure 6A,B). In the Kaplan–Meier analysis, high-level expression of YKL40 revealed a trend toward poorer OS and PFS, although the differences were not statistically significant (Figure 6C). Similarly, for ITGB4, high-level expression revealed a trend toward poorer OS, but neither OS nor PFS showed significant differences (Figure 6D). Examining the relationship between clinicopathological factors and the expression of YKL40 and ITGB4, this study revealed a significant positive correlation between ITGB4 expression and CD163; however, no other significant correlations were observed (Table 2). Considering that the insufficient results in terms of clinical indicators could be attributed to the small number of HGSOC cases, database analysis, including a larger number of HGSOC cases, was conducted. TNMplot revealed that, in serous ovarian cancer (SOC) tissues, both YKL40 and ITGB4 were significantly upregulated in tumor tissues compared with normal tissues (Figure 6E,G). Furthermore, the Kaplan–Meier plotter revealed that high-level expression of both YKL40 and ITGB4 was significantly associated with poorer PFS in SOC cases (Figure 6F,H).

## 3. Discussion

In this study, immunohistochemistry revealed that high-level infiltration of TAMs correlates with a poor prognosis in ovarian cancer. In vitro experiments demonstrated that the interaction between EOC cells and macrophages leads to the polarization of macrophages into TAMs. Furthermore, the secretion of YKL40 induced by the interaction between EOC cells and macrophages was shown to enhance the phosphorylation of Erk via ITGB4, thereby promoting the migration and invasion of EOC cells, suggesting its significant role in cancer progression (Figure 7). The importance of YKL40 and ITGB4 in the progression of EOC was supported by database analyses.

Previous studies have reported an association between a poor EOC prognosis and macrophage infiltration [20,21,22,23]. In HGSOC, increased CD68 expression is associated with a lower OS rate [24]. The expression levels of CD163, CD204, and CD206 were found to be significantly higher in HGSOC than in healthy controls [25]. In serous or mucinous ovarian tumors, CD204 levels significantly increase in malignant tumors compared with those in benign tumors [26]. This is consistent with our findings in HGSOC. Additionally, immunohistochemistry revealed a positive correlation between chemotherapy resistance and macrophage infiltration in the present study. The role of TAMs in chemoresistance has been observed in ovarian cancer, despite the relatively limited number of studies published on this topic. Zhu et al. have demonstrated that miR-223 is enriched in exosomes released from TAMs and transferred to co-cultured EOC cells under hypoxic conditions, which results in enhanced chemoresistance of EOC cells [27]. Although TAMs are associated with chemoresistance in many other types of cancer [28], this study did not extensively investigate the relationship between TAMs and chemoresistance.

In this study, the authors investigated the interaction between EOC cells and peripheral blood-derived macrophages using an indirect co-culture system established in the authors’ laboratory. Previous reports have indicated that macrophages contribute to ovarian cancer progression through various signaling pathways, including NF-κB [29], PI3K/Akt [30], and JAK/STAT [31,32]. In our study, activation of the Erk pathway was consistently observed across three EOC cell lines. Similarly, Zhang et al. have reported that a macrophage-conditioned medium activates the Erk pathway in SKOV3 cells, promoting cell proliferation and migration [33].

Herein, the authors identified humoral factors that may influence the malignant phenotypes of EOC cells, including OPN, MMP9, IL-17A, CCL2, and IL-8, in addition to YKL40, using a cytokine array. These factors are secreted by tumor cells, macrophages, and/or other immune cells, contributing to tumor progression in ovarian cancer [34,35,36,37,38]. The main reports of ovarian cancer are as follows. OPN promotes cell survival and increases HIF-1α expression through the PI3K/Akt pathway in cancer cells [39]. The matriptase-activated PAR-2/PI3K/Akt/MMP9 signaling axis results in the cleavage of E-cadherin and the release of soluble E-cadherin, disrupting cell–cell interactions and facilitating peritoneal dissemination in cancer cells [40]. In the direct co-culture supernatant of SKOV3- and THP-1-derived macrophages, IL-8 levels are significantly elevated, leading to macrophage polarization to the M2 phenotype and the induction of stem cell-like characteristics in cancer cells via STAT3 phosphorylation [41]. CCL2 produced by omental adipocytes binds to CCR2, activating the PI3K/Akt/mTOR pathway and its downstream effectors HIF-1α and VEGF-A, promoting migration and omental metastasis in cancer cells [42]. These reports support the finding that several cytokines that resulted from our cytokine array are crucial factors in the progression of ovarian cancer. Among these, there are few reports on the association between YKL40 and ovarian cancer, and its mechanism remains unclear, which brought about the focus on YKL40 in this study.

YKL40, also known as chitinase-3-like-protein-1, belongs to the glycoside hydrolase family 18 and plays significant roles in oxidative stress, apoptosis, fibrosis, and inflammation. YKL40 is overexpressed in various cancers, such as breast cancer, prostate cancer, colorectal cancer, and glioblastoma, and elevated levels of YKL40 are significantly associated with a poor prognosis [43]. The addition of YKL40 promotes the proliferation of HEK293 (human embryonic kidney cells) and U373 (human glioblastoma cells) cells through Erk phosphorylation [44] and the proliferation and migration of SW480 and COLO205 cells (colorectal cancer cells) via the NF-κB pathway [45]. Recent reports have indicated that YKL40 contributes to increased PD-L1 expression and immune suppression in gallbladder cancer and glioblastoma [46,47]. In vitro experiments on ovarian cancer have shown that YKL40 enhances the expression of stemness-related genes through the Erk and Akt pathways, increasing chemoresistance and the ability to form tumor spheroids [48]. The authors first reported that YKL40 secretion was induced through the interaction between macrophages and EOC cells and demonstrated its contribution to the malignant phenotypes of EOC cells via the Erk pathway.

YKL40 is known to be secreted by tumor cells and non-tumor cells (such as macrophages, neutrophils, stem cells, bone cells, fibroblast-like cells, endothelial cells, and vascular smooth muscle cells) [43]. Single-cell RNA sequencing has revealed significant YKL40 expression in M2 macrophages in gallbladder cancer [47], whereas in glioblastoma, YKL40 is highly expressed in tumor cells [46]. In pancreatic ductal adenocarcinoma, proteomics analysis revealed that YKL40 is abundantly present in the extracellular vesicles released by macrophages [49]. This study could not definitively determine whether YKL40 is secreted from EOC cells or macrophages. However, previous reports have suggested that it may be secreted by both cell types in ovarian cancer. At the very least, the fact that the interaction between EOC cells and macrophages enhances the secretion of YKL40 has been confirmed. Although not demonstrated in this study, the authors’ previous reports have shown that YKL40 promotes M2 polarization, proliferation, and migration of macrophages [19]. Although previous data did not mention the M2 subtype (e.g., M2a, M2b, M2c, and M2d), it is possible that it could be classified as M2c or M2d, characterized by high CD163 expression [50]. Zhao et al. have also demonstrated that YKL40 interacts with CD44 to promote M2 polarization of macrophages [46]. The interaction between macrophages and EOC cells leading to TAM polarization is supported by reports indicating that IL-4, IL-13, and microRNA secreted by EOC cells induce the M2 phenotype in TAMs [21,51,52]. YKL40 is one of the differentiation factors of TAMs. Thus, YKL40 is an important factor in the interaction between EOC cells and macrophages.

YKL40 is known to activate multiple signaling pathways by binding to IL13Rα2 and CD44 or by inducing the coordination of syndecan-1 with integrin α_v_β_3_ or integrin α_v_β_5_ [43]. The authors’ laboratory previously discovered ITGB4 as a novel receptor for YKL40 in esophageal epithelial cells, demonstrating molecular interactions through immunoprecipitation [19]. ITGB4 forms a heterodimer exclusively with integrin α6 [53]. To our knowledge, no studies have reported the relationship between YKL40 and ITGB4 in ovarian cancer. In this study, it was newly discovered that ITGB4-mediated Erk activation influences proliferation, migration, and invasion in EOC cells apart from SKOV3 cells. Previous reports have suggested that the p53 mutation may be related to the suppression of ITGB4 function [54,55]. Lee et al. have reported that anti-ITGB4 antibody does not inhibit adhesion in p53-null SKOV3 cells but significantly inhibits adhesion in SKOV3 cells transfected with p53^R248^ mutation and in OVCAR3 cells, which naturally possess the p53^R248^ mutation [55]. KURAMOCHI cells carry the p53^D281Y^ mutation, raising the intriguing possibility that these p53 mutations affect ITGB4 function. However, the role of p53 in ITGB4 function in ovarian cancer remains largely unexplored and requires further research.

No significant relationship was found between YKL40 and ITGB4 expression in HGSOC tissues and prognosis or clinicopathological factors. However, the database analysis revealed that YKL40 and ITGB4 expression was higher in ovarian cancer tissues than in normal tissues and that PFS was poorer in the groups with high-level expression of YKL40 and ITGB4. Høgdall et al. have also reported no correlation between YKL40 expression in EOC tissues and prognosis. Nonetheless, they found that, when compared with normal levels, elevated plasma YKL40 levels were significantly associated with decreased OS rates [56]. Furthermore, it has been shown that patients with stage III ovarian cancer and high plasma YKL40 levels have significantly decreased OS rates [57]. One study has evaluated YKL40 mRNA levels in EOC tissues and found that high-level expression of YKL40 was significantly associated with histological type (serous carcinoma), advanced stage, chemoresistance, recurrence, and poor OS and PFS [58]. Other studies have also reported the relationship between plasma YKL40 levels and chemoresistance [59,60]. YKL40 may not only predict prognosis and chemoresistance but also serve as a useful marker for the early detection of ovarian cancer. Dupont et al. assessed serum YKL40 levels in healthy individuals, those at a high risk of ovarian cancer, and those with EOC, finding that preoperative YKL40 levels were elevated in 72% of patients with ovarian cancer, outperforming CA125 levels, which were elevated in 46% [61]. However, to date, no evaluations of ITGB4 immunostaining in ovarian cancer have been reported. Given the heterogenous nature of ovarian cancer tissues and the fact that YKL40 is a secreted factor, comprehensive evaluations using plasma or mRNA derived from whole tissues rather than immunohistochemical evaluation using limited tissue samples may provide more accurate assessments.

This study has certain limitations. First, the sample size for the clinical analysis was small, which may have affected the assessment of YKL40 and ITGB4. Increasing the sample size could potentially alter the findings. Additionally, HGSOC tissues were obtained before chemotherapy, either during primary debulking surgery or diagnostic laparoscopy, resulting in inconsistent tissue samples from various locations, such as the ovaries, omentum, and peritoneum. This variability might have affected the accuracy of the tissue assessment. Second, in this study, in vitro experiments on chemoresistance associated with TAM markers could not be conducted. Future research needs to explore the impact of YKL40 on chemoresistance. Third, this study did not include in vivo experiments to validate the findings. Previous in vivo studies have demonstrated that injecting spheroids of OVCAR3 cells transfected with YKL40 into mice significantly increased tumor formation when compared with non-transfected cells [48]. Additionally, YKL40-neutralizing antibodies inhibit tumor progression in brain tumor or gallbladder cancer cells in immunodeficient mice [62,63]. Among reports concerning integrins, it has been demonstrated that an integrin α5β1-blocking antibody inhibits the growth and dissemination of ovarian cancer in xenograft models [10]. These findings suggest that the inhibition of YKL40 and ITGB4 may suppress tumor progression in ovarian cancer. Future research should investigate the inhibitory effects of YKL40 and ITGB4 on EOC progression using animal models.

## 4. Materials and Methods

### 4.1. Cell Lines and Cell Cultures

Three human EOC cell lines (KURAMOCHI, SKOV3, and OVCAR3) were obtained from the Japanese Collection of Research Bioresources Cell Bank (Osaka, Japan), American Type Culture Collection (Manassas, VA, USA), and Biological Resource Center (Tsukuba, Japan), respectively, and maintained in RPMI 1640 medium (Wako, Osaka, Japan) with 10% fetal bovine serum (FBS; Sigma-Aldrich, St. Louis, MO, USA) and 1% antibiotic/antimycotic stock solution (Wako) at 37 °C in a 5% CO_2_ atmosphere. Peripheral blood mononuclear cells (PBMCs) obtained from healthy volunteers were purified on autoMACS Pro Separator (Miltenyi Biotec, Bergisch Gladbach, Germany) using anti-CD14 microbeads (Miltenyi Biotec). For the differentiation of macrophages, PBMCs were incubated with 10 ng/mL macrophage colony-stimulating factor (R&D Systems, Minneapolis, MN, USA) and 1 ng/mL granulocyte-macrophage colony-stimulating factor (R&D Systems) for 6 days in RPMI 1640 medium (Wako) with 10% FBS and 1% antibiotic/antimycotic stock solution at 37 °C in a 5% CO_2_ atmosphere.

### 4.2. Indirect Co-Culture System

The macrophages were established in a 0.4 µm-pore insert (Falcon, BD, Franklin Lakes, NJ, USA) at a density of 2.0 × 10^5^ cells/well. One day before the co-culture, EOC cells were seeded in six-well plates at a density of 2.0 × 10^5^ cells/well in RPMI 1640 with 0.1% FBS. The macrophages in the inserts were washed three times with RPMI 1640 with 0.1% FBS. Both EOC cells and macrophages were co-cultured for 2 days.

### 4.3. Real-Time Quantitative Polymerase Chain Reaction

The RNA of the cultured cells was extracted using the RNeasy Mini Kit (Qiagen, Hilden, Germany) according to the manufacturer’s instructions and quantified using a NanoDrop Lite (Thermo Fisher Scientific, Waltham, MA, USA). The qPCR for CD163, CD204, ITGB4, and the control gene GAPDH were performed using SYBR Green PCR Master Mix (Applied Biosystems, Foster City, CA, USA). The threshold cycle (C_T_) values were determined by plotting the observed fluorescence against the cycle number. C_T_ values were analyzed using the comparative C_T_ method and normalized to those of GAPDH. The sequences of the primers are shown in Appendix A.

### 4.4. Western Blotting Analysis

Cells were lysed on ice with a lysis buffer (50 mM Tris-HCl at pH 7.5 with 125 mM NaCl, 5 mM EDTA, and 0.1% Triton X-100) containing 1% protease inhibitor and 1% phosphatase inhibitor cocktails (Sigma-Aldrich). The resulting lysates were separated on 5–20% sodium dodecyl sulfate-polyacrylamide gels and transferred to a membrane with an iBlot Gel Transfer Stack (Invitrogen, Carlsbad, CA, USA). The membrane was blocked with 5% skim milk and subsequently incubated with primary antibody overnight at 4 °C. After washing three times with TBST (Tris-HCl containing NaCl and Tween 20), the membranes were incubated at room temperature for 90 min with secondary antibody. After washing three times with TBST, the protein bands were detected using ImmunoStar Reagents (Wako). The primary antibodies were as follows: rabbit antibody against phosphorylated Erk1/2 (1:200; number 9101; Cell Signaling Technology [CST], Danvers, MA, USA), rabbit antibody against Erk1/2 (1:500; number 9102; CST), rabbit antibody against β-actin (1:2000; number 4970; CST), and rabbit antibody against ITGB4 (1:500; number 14803; CST). The secondary antibodies were HRP-conjugated anti-mouse and anti-rabbit IgG (1:1000; Cytiva, Marlborough, MA, USA). 

### 4.5. Cytokine Array Analysis

KURAMOCHI in six-well plates (2.0 × 10^5^ cells/well) and/or macrophages in 0.4 µm-pore inserts (2.0 × 10^5^ cells/well) were cultured in RPMI 1640 with 0.1% FBS for 48 h, and the supernatants of the KURAMOCHI and/or macrophages from each monoculture or co-culture were subsequently collected. According to the manufacturer’s instructions, the supernatants were applied to the Human XL Cytokine Array Kit (ARY022B; R&D systems).

### 4.6. Enzyme-Linked Immunosorbent Assay

For SKOV3 and OVCAR3, the supernatants were collected in the same way as that for KURAMOCHI, as described above. The YKL40 concentrations in the supernatants were analyzed using a Human Chitinase 3-like 1 Quantikine ELISA kit (DC3L 10; R&D Systems) following the manufacturer’s instructions. The optical density of each sample was measured at 450 and 570 nm using a microplate reader (Infinite 200 PRO; Tecan, Mannedorf, Switzerland) and the YKL40 concentrations were calculated from the absorbance values using a standard curve.

### 4.7. ITGB4 Knockdown in EOC Cell Lines Using Small Interfering RNA

EOC cells were transfected with 20 nM small interfering RNA (siRNA) targeting ITGB4 (siITGB4; Santa Cruz Biotechnology, Dallas, TX, USA) or 20 nM negative control siRNA (siNC; Sigma-Aldrich) for 72 h using Lipofectamine RNAiMAX (Invitrogen).

### 4.8. Transwell Proliferation Assay

PBMCs (3.0 × 10^4^ cells/well) were differentiated into macrophages in 0.4 µm-pore 24-well inserts. On day 6, polarized macrophages were washed with RPMI 1640 with 0.1% FBS three times and subsequently co-cultured with EOC cells (4.0 × 10^4^ cells/well) in 24-well plates in RPMI 1640 with 0.1% FBS. After 48 h, the CellTiter 96 AQ_ueous_ One Solution Reagent (Promega, Madison, WI, USA) was added. Absorbance was measured at 492 nm using a microplate reader (Infinite 200 PRO; Tecan). In some experiments, rhYKL40 (R&D Systems) (250 ng/mL) with or without MEK1/2 inhibitor (PD98059; 10 µM; CST) was added to the culture medium when EOC cells (5.0 × 10^3^ cells/well) were seeded into 96-well plates.

### 4.9. Transwell Migration/Invasion Assay

PBMCs (1.0 × 10^5^ cells/well) were differentiated into macrophages in 24-well plates. On day 6, macrophages were washed with RPMI 1640 with 0.1% FBS three times and subsequently co-cultured with EOC cells (1.0 × 10^5^ cells/well for transwell migration assay or 2.0 × 10^5^ cells/well for transwell invasion assay) seeded in 8 µm-pore inserts (Falcon, BD, Franklin Lakes, NJ, USA) or 8 µm-pore inserts coated with Matrigel (Corning, Corning, NY, USA) in RPMI 1640 with 0.1% FBS. After 48 h, migrated or invaded EOC cells on the lower surface of the membranes were stained using a Diff-Quik kit (Sysmex, Kobe, Japan). Five images at 200× magnification were obtained for each membrane using a charge-coupled device camera (Olympus, Tokyo, Japan), and the cells were counted. In some experiments, rhYKL40 (250 ng/mL) with or without MEK1/2 inhibitor (PD98059; 10 µM; CST) was added to the culture medium in the lower chamber.

### 4.10. Tissue Samples

The tissue samples of patients with HGSOC who had undergone surgical resection from 2009 to 2019 at Kobe University Hospital (Kobe, Japan) were retrospectively collected. Patients with HGSOC who had received neoadjuvant therapy (chemotherapy or radiotherapy) before the resection were excluded. Tissue samples were obtained through diagnostic laparoscopy or primary debulking surgery. Finally, tissue samples from 65 patients with HGSOC were collected, comprising 40 ovarian samples, 13 peritoneal samples, five omental samples, four fallopian tube samples, and three samples from other sites (including umbilical tumors, small intestinal tumors, and inguinal lymph nodes). Clinicopathological and histological factors were classified according to the International Federation of Gynecology and Obstetrics staging system and TNM classification [64]. Informed consent was obtained from all the patients for the use of their tissue samples and clinical data. The Institutional Review Board of Kobe University (B240008) approved all the study protocols, and the study was conducted in accordance with the guidelines of the 1964 Declaration of Helsinki and its later amendments.

### 4.11. Immunohistochemistry

Immunohistochemistry was performed on 4 µm-thick sections of paraffin-embedded human HGSOC tissues using the Leica BOND-MAX automated system and BOND Polymer Refine Detection kit (Leica Biosystems, Bannockburn, IL, USA). The sections were deparaffinized and heated in a sodium citrate buffer (pH 6.0) at 100 °C for 20 min to expose antigens and subsequently incubated with primary antibodies for 15 min, post-primary solution for 8 min, peroxide block for 5 min, 3,3′-diaminobenzidine as chromogenic substrate for 10 min, and hematoxylin counterstaining for 5 min. The antibodies used for antigen detection in the tissue samples were as follows: mouse antibody against CD68 (1:300; KP-1; Leica Biosystems), CD163 (1:400; NCL-L-CD163; Novocastra, Newcastle upon Tyne, UK), and CD204 (1:200; KT022; TransGenic, Kobe, Japan), and rabbit antibody against YKL40 (1:500; ab77528; abcam, Cambridge, UK) and ITGB4 (1:200; 14803; CST).

To quantify CD68^+^, CD163^+^, and CD204^+^ cells, intra-tumoral regions were evaluated. High-power field images (three per sample) were analyzed, and the samples were classified into two groups (high or low) based on the median of the three images. YKL40 expression was assessed by the staining intensity within the YKL40-positive areas of the entire tumor in high-power fields. The staining intensity was scored as follows: 0 for negative, 1+ for mild, 2+ for moderate, and 3+ for strong. A score of 0 was defined as a low-level expression, whereas scores of 1, 2 and 3 were considered high-level expressions. ITGB4 expression was evaluated based on the percentage of ITGB4-positive areas within the entire tumor in high-power fields, excluding staining intensity from the assessment. The percentage of ITGB4-positive areas was scored based on five grades: 0 (≤1%), 1 (>1% and ≤5%), 2 (>5% and ≤30%), 3 (>30% and ≤50%), and 4 (>50%). Scores of 0, 1 and 2 were defined as low-level expressions, and scores of 3 and 4 were considered high-level expressions.

### 4.12. Statistical Analysis

All in vitro experiments were conducted in three independent replicates, each performed three separate times. The in vitro experimental data are expressed as mean ± standard error, and significance was assessed using two-sided *t*-tests. *χ*^2^ tests were used to estimate the relationship between the immunohistochemical results and clinicopathological factors. The Kaplan–Meier method was used to create curves for OS and PFS, and the differences were evaluated using the log-rank test. *p* < 0.05 was considered statistically significant. Statistical analyses were performed using R software version 4.1.2 (R Foundation for Statistical Computing, Vienna, Austria).

## 5. Conclusions

This study revealed that YKL40, of which the secretion was enhanced through the interaction between cancer cells and macrophages, is involved in EOC progression by promoting the proliferation, migration, and invasion of cancer cells via ITGB4. Therefore, YKL40 holds potential as a novel therapeutic target for EOC progression. The findings of this study may contribute to the prognostication and treatment of ovarian cancer.

## Figures and Tables

**Figure 1 ijms-25-10598-f001:**
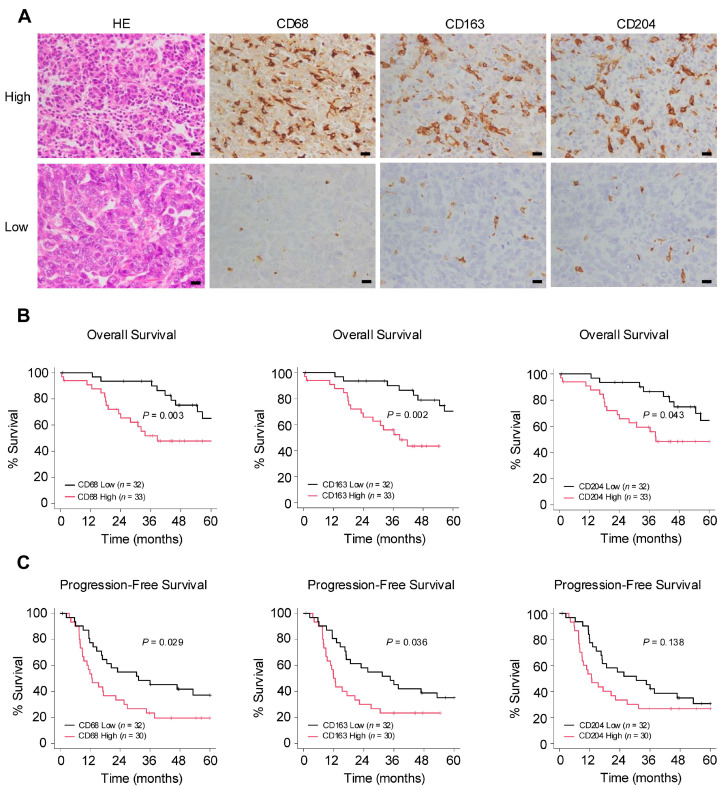
Number of infiltrating macrophages correlates with poor prognosis in ovarian cancer. (**A**): Immunohistochemistry for the detection of pan-macrophage marker CD68 and tumor-associated macrophage (TAM) markers CD163 and CD204 was performed in 65 human high-grade serous ovarian carcinoma (HGSOC) tissues, and the respective positive cells were counted in high-power fields (three images/case). Patients were subsequently categorized into high or low groups based on the median of the average count of infiltrating macrophages. (**B**,**C**): Kaplan–Meier analysis of (**B**) overall survival (OS) in 65 patients with HGSOC with low (*n* = 32) and high (*n* = 33) counts and of (**C**) progression-free survival (PFS) in 62 patients with HGSOC with low (*n* = 32) and high (*n* = 30) counts, based on low- and high-level CD68, CD163 and CD204 expression. *p*-values were determined using the log-rank test. Scale bar: 20 µm (**A**).

**Figure 2 ijms-25-10598-f002:**
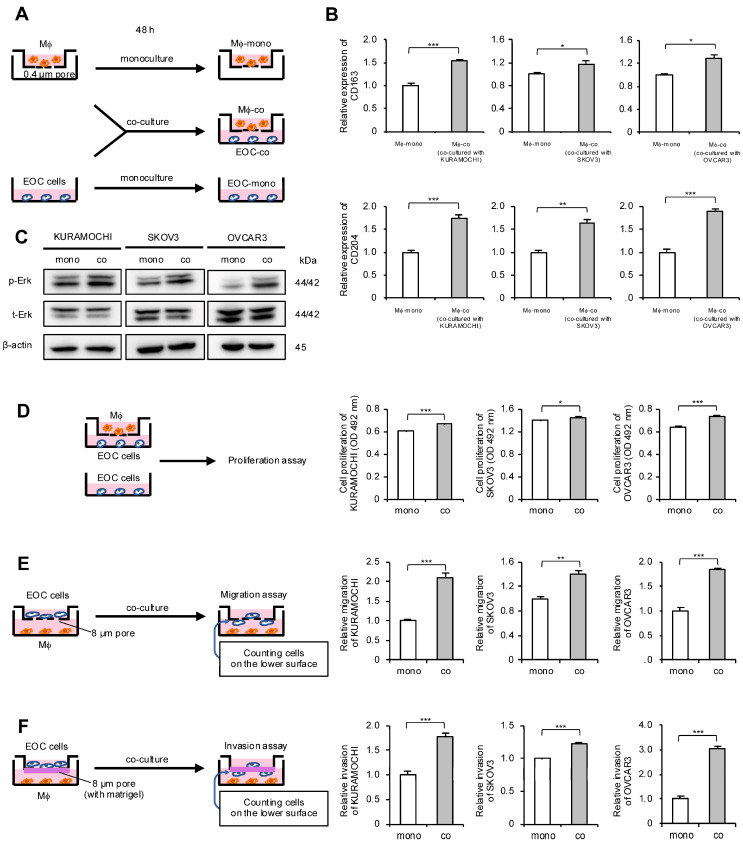
Co-culture of macrophages and epithelial ovarian cancer (EOC) cells promotes polarization of macrophages into TAMs and enhances malignant phenotypes of EOC cells. (**A**): Indirect co-culture of EOC cells (KURAMOCHI, SKOV3 and OVCAR3) in the lower chamber and macrophages in the transwell assay with 0.4 µm pores was performed for 48 h. For comparison, EOC cells and macrophages were also cultured alone for 48 h as monocultured controls. (**B**): Expression of TAM markers CD163 and CD204 was compared using real-time quantitative PCR (qPCR) between monocultured and co-cultured macrophages. (**C**): Expression of extracellular signal-regulated kinase (Erk) and phosphorylated Erk (p-Erk; Thr202/Tyr204) in monocultured and co-cultured EOC cells was evaluated using Western blotting. β-actin was used as a control. (**D**): MTS assays were performed to compare the proliferation of monocultured and co-cultured EOC cells. (**E**): Transwell migration assays were performed to compare monocultured and co-cultured EOC cells. After 48 h of incubation, migrating cells into the lower surface were counted in five random fields per chamber. Representative images are shown in Appendix A. (**F**): Transwell invasion assays were performed to compare monocultured and co-cultured EOC cells. After 48 h of incubation, invading cells into the lower surface were counted in five random fields per chamber. Representative images are shown in Appendix A. Mϕ, macrophages. * *p* < 0.05, ** *p* < 0.01, *** *p* < 0.001.

**Figure 3 ijms-25-10598-f003:**
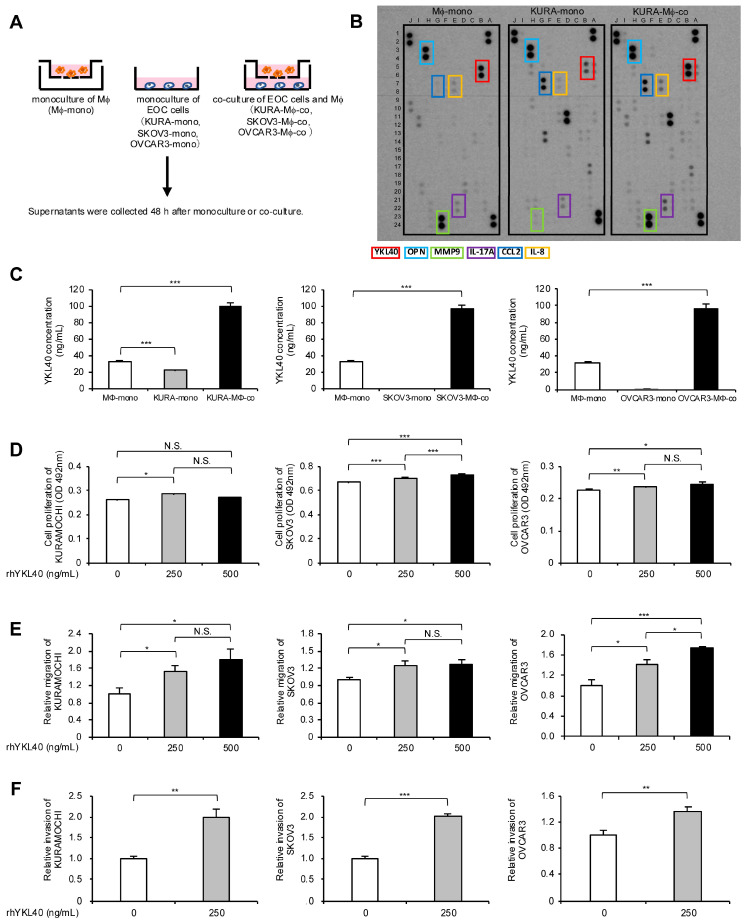
Increased YKL40 resulting from indirect co-culture of macrophages and EOC cells promotes malignant phenotypes of EOC cells. (**A**): Macrophages and EOC cells were each monocultured for 48 h, and macrophages and EOC cells were co-cultured for 48 h. Cell supernatants were collected. (**B**): Cytokine arrays were performed on each collected cell supernatant. Colored boxes indicate spots of enhanced signal in the supernatants of the co-culture compared with the KURAMOCHI and macrophage monocultures. (**C**): ELISA was performed to investigate the secretion of YKL40 in three types of cell supernatants for each EOC cell line. (**D**–**F)**: Each EOC cell line was treated with recombinant human YKL40 (rhYKL40) at concentrations of 0, 250 and 500 ng/mL, which was performed for (**D**) MTS assays, (**E**) transwell migration assays, and (**F**) transwell invasion assays after 48 h. (**E**) Migrating cells and (**F**) invading cells were counted in five random fields per chamber. Representative images of transwell migration and invasion assays are shown in Appendix A, respectively. * *p* < 0.05, ** *p* < 0.01, *** *p* < 0.001. N.S., not significant.

**Figure 4 ijms-25-10598-f004:**
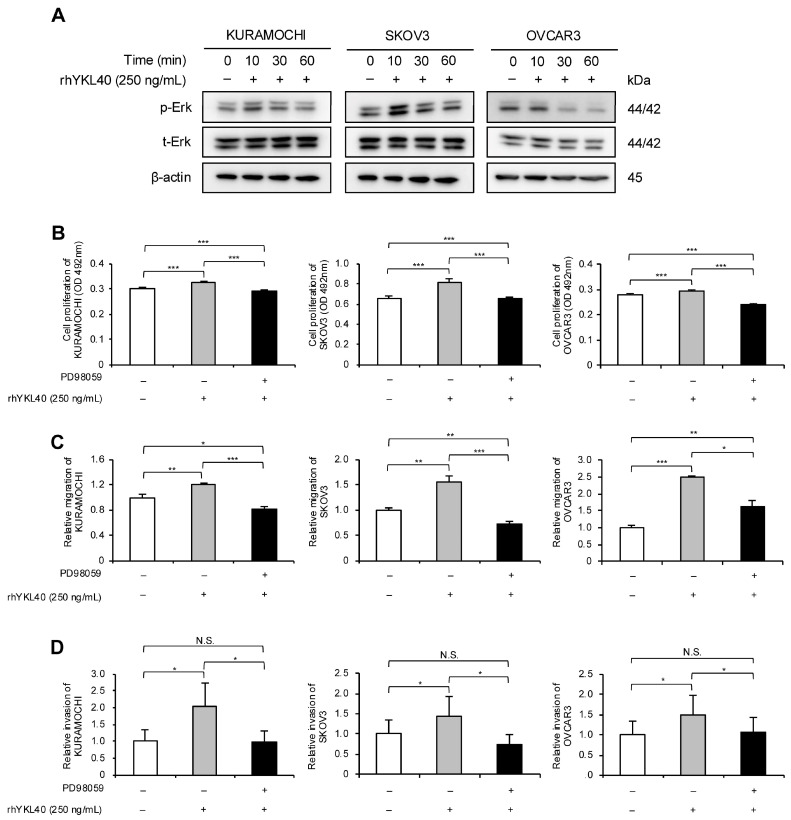
YKL40 promotes malignant phenotypes of EOC cells via the Erk pathway. (**A**): Time-dependent expression levels of Erk and p-Erk in EOC cells treated with rhYKL40 (250 ng/mL) were analyzed using Western blotting, with β-actin as control. (**B**–**D**): Changes in (**B**) proliferation, (**C**) migration and (**D**) invasion of EOC cells following rhYKL40 (250 ng/mL) treatment with or without MEK1/2 inhibitor (PD98059; 10 µM) were assessed using (**B**) MTS assay, (**C**) transwell migration assay, and (**D**) transwell invasion assay, respectively. Representative images of transwell migration and invasion assays are shown in Appendix A, respectively. * *p* < 0.05, ** *p* < 0.01, *** *p* < 0.001. N.S., not significant. Furthermore, the original Western blotting images are provided as Appendix A.

**Figure 5 ijms-25-10598-f005:**
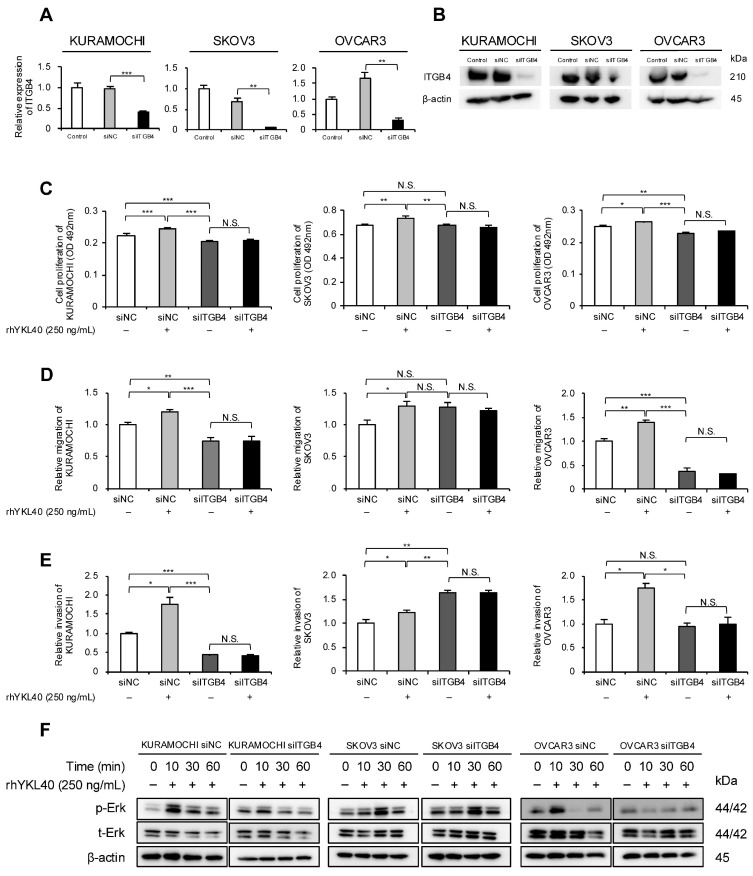
Enhanced effects of malignant phenotypes of EOC cells by YKL40 mediated through integrin β4 (ITGB4). (**A**,**B**): EOC cells were transfected with siRNA targeting ITGB4 (siITGB4, 20 nM). The knockdown efficiency of ITGB4 in EOC cells was assessed using (**A**) qPCR and (**B**) Western blotting. Negative control siRNA (siNC, 20 nM) was used as a negative control. (**C**–**E**): EOC cells transfected with either siITGB4 or siNC were treated with or without rhYKL40 (250 ng/mL), and the proliferation, migration, and invasion were compared using (**C**) MTS assay, (**D**) transwell migration assay, and (**E**) transwell invasion assay, respectively. Representative images of transwell migration and invasion assays are shown in Appendix A, respectively. (**F**): EOC cells transfected with either siNC or siITGB4 were treated with rhYKL40 (250 ng/mL) and time-dependent expression levels of Erk and p-Erk were analyzed using Western blotting, with β-actin as control. * *p* < 0.05, ** *p* < 0.01, *** *p* < 0.001. N.S., not significant.

**Figure 6 ijms-25-10598-f006:**
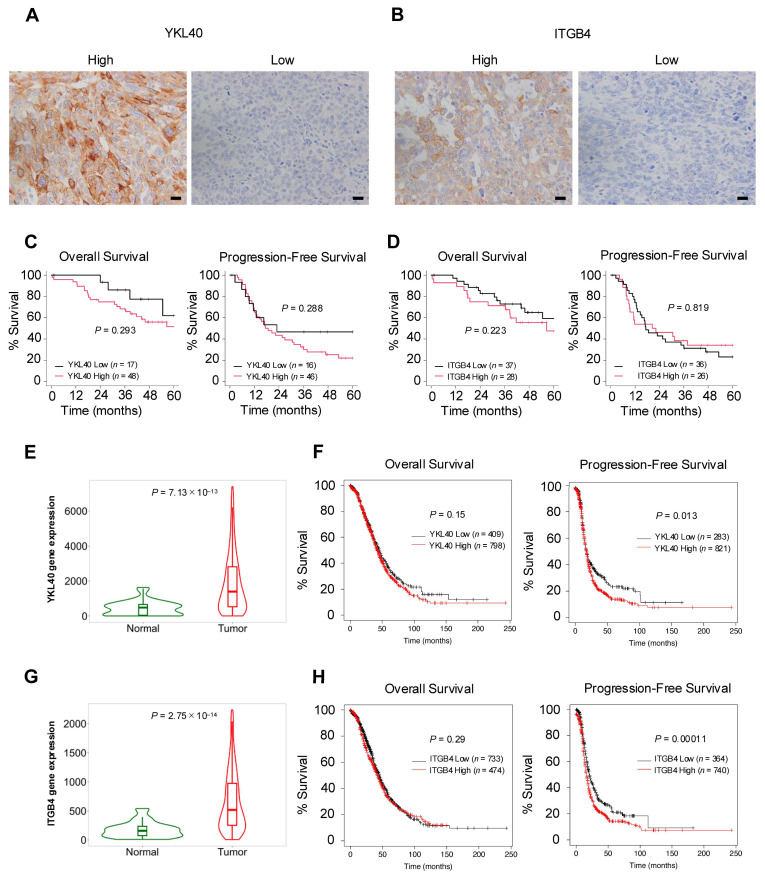
The YKL40/ITGB4 axis may act as an indicator of poor prognosis in EOC. (**A**,**B**): Immunohistochemistry for (**A**) YKL40 and (**B**) ITGB4 was performed in 65 human HGSOC tissues, and respective positive cells were evaluated in high-power fields (three images/case). Patients were subsequently categorized into high and low groups. (**C**): Kaplan–Meier analysis of OS in 65 patients with HGSOC with low (*n* = 17) and high (*n* = 48) expression and PFS in 62 patients with HGSOC with low (*n* = 16) and high (*n* = 46) expression, based on low- and high-level YKL40 expression. (**D**): Kaplan–Meier analysis of OS in 65 patients with HGSOC with low (*n* = 37) and high (*n* = 28) expression and PFS in 62 patients with HGSOC with low (*n* = 36) and high (*n* = 26) expression, based on low- and high-level ITGB4 expression. *p*-values were determined using the log-rank test. (**E**–**H**): Database analyses for YKL40 and ITGB4 were performed using TNMplot and Kaplan–Meier plotter. TNMplot was used to evaluate the difference in expression levels of (**E**) YKL40 between normal and tumor tissues in EOC tissues. Kaplan–Meier plotter was used to assess the prognosis of OS and PFS by comparing low- and high-level expression groups of (**F**) YKL40. TNMplot was used to evaluate the difference in expression levels of (**G**) ITGB4 between normal and tumor tissues in EOC tissues. Kaplan–Meier plotter was used to assess the prognosis of OS and PFS by comparing low- and high-level expression groups of (**H**) ITGB4. Scale bar: 20 µm (**A**,**B**).

**Figure 7 ijms-25-10598-f007:**
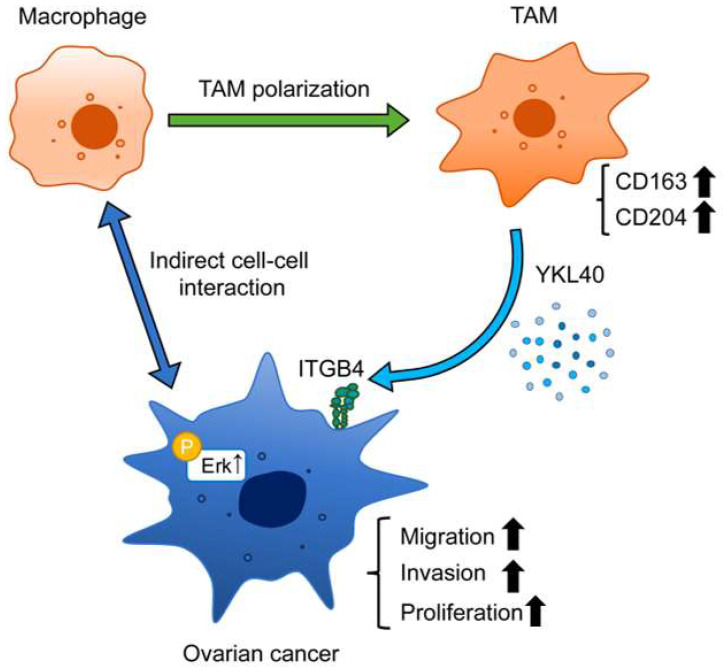
Schematic diagram illustrating the role of YKL40 in the EOC microenvironment. Indirect interactions with EOC cells lead to polarization of macrophages into TAMs. YKL40 is mainly secreted by TAMs. This activates the Erk pathway via ITGB4, promoting the migration and invasion of EOC cells.

**Table 1 ijms-25-10598-t001:** Relationship between expression levels of macrophage markers and clinicopathological factors.

Clinicopathological Factor		CD68 ^†^			CD163 ^†^			CD204 ^†^		
		Low (*n* = 32)	High (*n* = 33)	*p*-Value	Low (*n* = 32)	High (*n* = 33)	*p*-Value	Low (*n* = 32)	High (*n* = 33)	*p*-Value
Age	<65 years	16	14	0.622	16	14	0.622	16	14	0.622
	≥65 years	16	19		16	19		16	19	
Stage ^‡^	Ⅰ + Ⅱ	10	6	0.260	8	8	1.000	9	7	0.574
	Ⅲ + Ⅳ	22	27		24	25		23	26	
Chemoresistance ^§^	+	5	14	0.011 *	5	14	0.025 *	5	14	0.025 *
	−	26	14		24	16		24	16	
Lymph node metastasis ^¶^	+	11	20	0.048 *	12	19	0.138	13	18	0.324
	−	21	13		20	14		19	15	
Peritoneal dissemination	+	22	27	0.260	24	25	1.000	23	26	0.574
	−	10	6		8	8		9	7	

Data were analyzed using a χ^2^-test. *p* < 0.05 was considered statistically significant: * *p* < 0.05; ^†^ classified into two groups, namely, the high-infiltration group (High) and low-infiltration group (Low), based on the median of the average count of infiltrating macrophages as determined by immunohistochemistry; ^‡^ referring to the 2014 FIGO classification for cancer of the ovary, fallopian tube, and peritoneum; ^§^ chemoresistance was defined as recurrence within less than 6 months. Patients that did not undergo chemotherapy were excluded; ^¶^ lymph node metastasis was analyzed using both pathological and clinical stages.

**Table 2 ijms-25-10598-t002:** Relationship of YKL40 and ITGB4 expression levels with clinicopathological factors.

Clinicopathological Factors		YKL40 ^†^			ITGB4 ^‡^		
		Low (*n* = 17)	High (*n* = 48)	*p*-Value	Low (*n* = 37)	High (*n* = 28)	*p*-Value
Age	<65 years	8	22	1.000	15	15	0.326
	≥65 years	9	26		22	13	
Stage ^§^	Ⅰ + Ⅱ	4	12	1.000	10	6	0.773
	Ⅲ + Ⅳ	13	36		27	22	
Chemoresistance ^¶^	+	6	13	0.345	26	11	0.158
	−	8	32		8	14	
Lymph node metastasis ^||^	+	7	24	0.583	19	12	0.617
	−	10	24		18	16	
CD68	High	6	27	0.166	17	17	0.317
	Low	11	21		20	11	
CD163	High	7	26	0.408	14	18	0.046 *
	Low	10	22		23	10	
CD204	High	8	25	0.783	15	18	0.080
	Low	9	23		22	10	

Data were analyzed using a χ^2^-test. *p* < 0.05 was considered statistically significant: * *p* < 0.05; **^†^** classified into two groups, namely, the high-intensity group (High) and low-intensity group (Low), based on the immunohistochemical intensity of YKL40 staining in the tumor; ^‡^ classified into two groups, namely, the high-extent group (High) and low-extent group (Low), based on the immunohistochemical extent of ITGB4 staining in the tumor; ^§^ referring to the 2014 FIGO classification for cancer of the ovary, fallopian tube, and peritoneum; ^¶^ chemoresistance was defined as recurrence within less than 6 months. Patients who did not undergo chemotherapy were excluded; ^||^ lymph node metastasis was analyzed using both pathological and clinical stages.

## Data Availability

The data presented in this study are available on request from the corresponding author.

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
