# Peer review of "YKL40/Integrin β4 Axis Induced by the Interaction between Cancer Cells and Tumor-Associated Macrophages Is Involved in the Progression of High-Grade Serous Ovarian Carcinoma"

_ijms, 2024, doi:10.3390/ijms251910598_

Round 1

Reviewer 1 Report

Comments and Suggestions for Authors

In this study authors evaluated the mechanisms related to tumor–stromal interactions in High-grade serous ovarian carcinoma (HGSOC) progression and found that patients with a higher infiltration of CD68+, CD163+, and CD204+ macrophages had a poorer prognosis. Moreover, co-culture of macrophages and EOC cells showed increased expression of the TAM markers CD163 and CD204, and enhanced proliferation, migration, and invasion. Furthermore, there was a higher YKL40 secretion  which increased proliferation, migration, and invasion via extracellular signal–regulated kinase (Erk) signaling. Authors also found that knockdown of integrin β4 suppressed YKL40–induced proliferation, migration, and invasion as well as Erk phosphorylation in some EOC cells. 

Although the manuscript is interesting and generally well written. Several points deserve to be improved. In particular:

Lines 45-52: it deserves to be pointed out that the high mortality rate is also due to chemoresistance occurrence (see PMID: 38203758 ).

Lines 57-60: it is important to highlight that stromal cells also regulate tumor angiogenesis (PMID: 37443812 ).

Figure 1A and 6A: Higher magnifications are necessary to appreciate tissue morphology and staining 

Table 1 and 2: Statistical significant differences shuld be highlighted in bold

Figure 2C, 4A, 5F: Densitometric analysis of pERK/ERK must be reported

Figure 2, 3,4, 5: Unless replaced with bigger images, invasion images at bottom of the graphs are useless.

Figure 5B: Densitometric analysis of ITGB4 must be reported

Authors must report the number of replicates (N) in the figures legends 

4.3. Real-time Quantitative Polymerase Chain Reaction: primers sequences should be insert in a dedicate table 

Reviewer 2 Report

Comments and Suggestions for Authors

For Figure 4a, the actin image on the left appears slightly off. The authors should recheck the actin staining for consistency and accuracy, ensuring the image doesn’t have artifacts. If necessary, they may need to reprocess or recapture the data to correct any discrepancies and make the image clearer.

In Figure 5E, the migration data needs better representation. The authors should adjust the plotting to make the data more discernible and visually coherent, possibly increasing resolution or rearranging elements for a cleaner view.

For Figure 4D, it is currently too small to be visible. The authors should enlarge this portion of the figure, adjusting the arrangement to ensure all parts are clearly presented and readable, improving overall figure quality.

Regarding Figure 5F, a semi-quantitative analysis should be conducted. The authors can achieve this using image analysis tools like ImageJ, ensuring the data is represented with appropriate statistical measures like mean and standard deviation, then visualizing the analysis in a clear graphical form.

Figure 6A should be aligned vertically with B and C for consistency in the visual flow of data. This will improve the coherence and readability of the figure arrangement.

For Figure 7, the authors should refine the appearance of the graphical elements and data points. Ensuring the labels for monocytes (MO) and tumor-associated macrophages (TAMs) are clear, and the arrows or annotations are pointing to the correct locations, possibly clarifying what “2?” represents. They should also mention the software used for plotting (such as GraphPad or MATLAB) to ensure reproducibility.

Lastly, the authors should consider addressing whether YPL4 shows selectivity toward specific subtypes of M2 macrophages (e.g., M2a, M2b, M2c). If they have evidence or experimental data supporting the selectivity, they should include that information to clarify the role of YPL4 in targeting these subtypes.

Comments on the Quality of English Language

good

Round 2

Reviewer 1 Report

Comments and Suggestions for Authors

the manuscript can be accepted in the present form

Reviewer 2 Report

Comments and Suggestions for Authors

Thank you for the revision.